PLOS **GLOBAL PUBLIC HEALTH**

# Variation in the incidence of type 1 diabetes mellitus in children and adolescents by world region and country income group: A scoping review

**Apoorva Gomber** [1]*, **Zachary J. Ward** [2], **Carlo Ross** [1,3], **Maira Owais** [1,4], **Carol Mita** [5], **Jennifer M. Yeh** [6], **Ché L. Reddy** [1], **Rifat Atun** [1,7]

1 Department of Global Health and Population, Harvard T.H. Chan School of Public Health, Boston, Massachusetts, United States of America, 2 Division of General Pediatrics, Boston Children's Hospital, Harvard Medical School, Boston, Massachusetts, United States of America, 3 Department of Global Health and Social Medicine, Harvard Medical School, Boston, Massachusetts, United States of America, 4 Center for Health Decision, Harvard T.H. Chan School of Public Health, Boston, Massachusetts, United States of America, 5 Manchester University NHS Foundation Trust, Manchester, United Kingdom, 6 Department of Biology, Department of Economics, Amherst College, Amherst, Massachusetts, United States of America, 7 Countway Library, Harvard Medical School, Boston, Massachusetts, United States of America

* apoorvagomber@hsph.harvard.edu

**Data Availability Statement:** All data are in the manuscript and/or Supporting information files.

## Abstract

### Introduction

Around 18.7 million of the 537 million people with diabetes worldwide live in low-income and middle-income countries (LMIC), where there is also an increase in the number of children, adolescents, and young adults diagnosed with type 1 diabetes (T1D). There are substantial gaps in data in the current understanding of the epidemiological patterns and trends in incidence rates of T1D at the global level.

### Methods

We performed a scoping review of published studies that established the incidence of T1D in children, adolescents, and young adults aged 0–25 years at national and sub-national levels using PubMed, Embase and Global Health. Data was analyzed using R programming.

### Results

The scoping review identified 237 studies which included T1D incidence estimates from 92 countries, revealing substantial variability in the annual incidence of T1D by age, geographic region, and country-income classification. Highest rates were reported in the 5–9 and 10–14 year age groups than in the 0–4 and 15–19 year age groups, respectively. In the 0–14 year age group, the highest incidence was reported in Northern Europe (23.96 per 100,000), Australia/New Zealand (22.8 per 100,000), and Northern America (18.02 per 100,000), while the lowest was observed in Melanesia, Western Africa, and South America (all < 1 per 100,000). For the 0–19 year age group, the highest incidence was reported in Northern Europe (39.0 per 100,000), Northern America (20.07 per 100,000), and Northern Africa

**Funding:** The authors received no specific funding for this work.

**Competing interests:** The authors have declared that no competing interests exist.

(10.1 per 100,000), while the lowest was observed in Eastern and Western Africa (< 2 per 100,000). Higher incidence rates were observed in high-income countries compared to LMICs. There was a paucity of published studies focusing on determining the incidence of T1D in LMICs.

## Conclusion

The review reveals substantial variability in incidence rates of T1D by geographic region, country income group, and age. There is a dearth of information on T1D in LMICs, particularly in sub-Saharan Africa, where incidence remains largely unknown. Investment in population-based registries and longitudinal cohort studies could help improve the current understanding of the epidemiological trends and help inform health policy, resource allocation, and targeted interventions to enhance access to effective, efficient, equitable, and responsive healthcare services.

## Introduction

Diabetes is a major threat to health systems globally. An estimated 537 million people are living with diabetes worldwide in 2021, with around 18.7 million people living in low-income and middle-income countries (LMICs) [1]. The number of children, adolescents, and young adults diagnosed with type 1 diabetes (T1D) in LMICs is rising with over 1.2 million estimated in 2021, with several countries reporting higher incidence rates than current available best estimates resulting in more than half (54%) of them <15 years of age [1].

The number of people living with diabetes is projected to increase to 643 million by 2030 and 783 million by 2045, with adverse consequences for the health and economic wellbeing of individuals, households, and countries [1–3]. An estimated 240 million people are presently living with undiagnosed diabetes worldwide with the global economic burden of diabetes estimated to increase from US$1.3 trillion in 2015 to $2.5 trillion by 2030 due to premature disability and mortality [4].

However, there are critical gaps in the current understanding of diabetes in children, adolescents, and young adults, with recent studies suggesting substantial variation in incidence at the global and sub-national levels, with higher incidence rates observed in several LMICs than earlier estimates indicated [5–11]. For many countries like Nigeria and South Africa, data are not available and are extrapolated from a nearby country with similar characteristics which may not be accurately presented. Less is known about the natural history, etiology, likelihood of complications, and the capability of health systems to deliver effective healthcare services to patients living with T1D in varied settings. It is crucial to understand the disease burden especially in LMICs where data is scant—and how it affects *Early Child Development* (ECD) and influences the active participation of adolescents and young adults in their economic and social pursuits. Such knowledge could inform health policies, resource allocation, and clinical practice to improve health outcomes for children, adolescents, and young adults in LMICs such as the need to secure adequate sustainable supplies of insulin.

There have been several notable initiatives to estimate the incidence of T1D documenting geographic and temporal trends, the impacts of health system barriers to diagnosis have not been quantified when estimating incidence rates, which may explain in part the large variation and increasing trends in observed incidence. In 1990, the World Health Organization

launched the Multinational Project for Childhood Diabetes (WHO DIAMOND Project) to determine the global incidence of T1D over ten years, from 1990 to 1999. The goals of this project were threefold, namely to: I) gather standard information on incidence, risk factors, complications, and mortality associated with type 1 diabetes; II) assess the efficiency and effectiveness of health care and the economics of diabetes; and III) establish national and international training programs in diabetes epidemiology [6]. The DIAMOND project demonstrated more than 350-fold variation in the incidence of T1D among countries, although incidence data from several LMICs were only available for the first five years of the study period. However, the WHO DIAMOND project played a significant catalytic role to inspire subsequent initiatives to develop regional, national and multi-country registries to monitor the incidence and prevalence of T1D worldwide [6, 9–12]. The International Diabetes Federation (IDF) Atlas 10th Edition provides the most current estimates for incidence rates of T1D from 215 countries and territories, grouped into seven IDF Regions: Africa (AFR), Europe (EUR), Middle East, and North Africa (MENA), North America and Caribbean (NAC), South and Central America (SACA), South-East Asia (SEA) and the Western Pacific (WP) which include newly available data from AFR since 2019. Previous estimates in the 9th Atlas from 2019 included 138 out of the 211 countries worldwide. Incidence rates were available for approximately 6% (3/47) of countries in sub-Saharan Africa, 31% in Western Pacific (11/36), 33% in North America and the Caribbean (8/24), 57% in the Middle East and North Africa (12/21), 57% in South-East Asia (4/7), 63% in South and Central America (12/19) and 77% (44/57) in Europe [8]. In 2021 IDF Atlas revision, only 97 of the 215 countries and territories have their own incidence data estimating a total of 1,211,900 children and adolescents under 20 years with T1D and 651,700 children under 15 years age. It is estimated that each year around 108,200 children and adolescents under 15 years are diagnosed with T1D with EUR and NAC regions reporting the highest number of prevalent cases and high incidence rates [1].

An initial search for systematic and scoping reviews on the global incidence of T1D in children, adolescents, and young adults was conducted, but no studies were identified. This study reviews the literature to examine the incidence of T1D in children, adolescents, and young adults (0–25 years) worldwide and explains the wide heterogeneity in the incidence rates that differ by age, sex, world region, and country income classification.

## Methods

### Search strategy

In designing and conducting this scoping review, we adopted the methodological framework proposed by Arksey and O'Malley [13]. A systematic scoping review was conducted according to the PRISMA recommendations retrieved from original papers published in English up to April 12, 2021, in peer-reviewed journals reporting the incidence of T1D among individuals aged <25 years of age in population-based studies and reporting the diagnostic criteria used to define T1D. The databases used for the literature search were Medline / PubMed (National Library of Medicine, NCBI), Embase (Elsevier, embase.com), and Global Health (C.A.B. International, Ebsco). The protocol for the search was registered in the International Prospective Register of Systematic Reviews (PROSPERO). Controlled vocabulary terms (i.e. MeSH, Emtree, CAB Thesaurus) were included when available and appropriate. The search strategies were designed and executed by a librarian (CM) from Harvard University in discussion with the authors. Tables A and B in S1 Text outline both the concepts and controlled vocabulary used for the search strategy and summary. Fig 1 presents the flow diagram of the bibliographic search using the PRISMA 2020 checklist (PRISMA-ScR checklist shown in S1 Text). Ethics approval for performing the study was not required as no primary data were included.

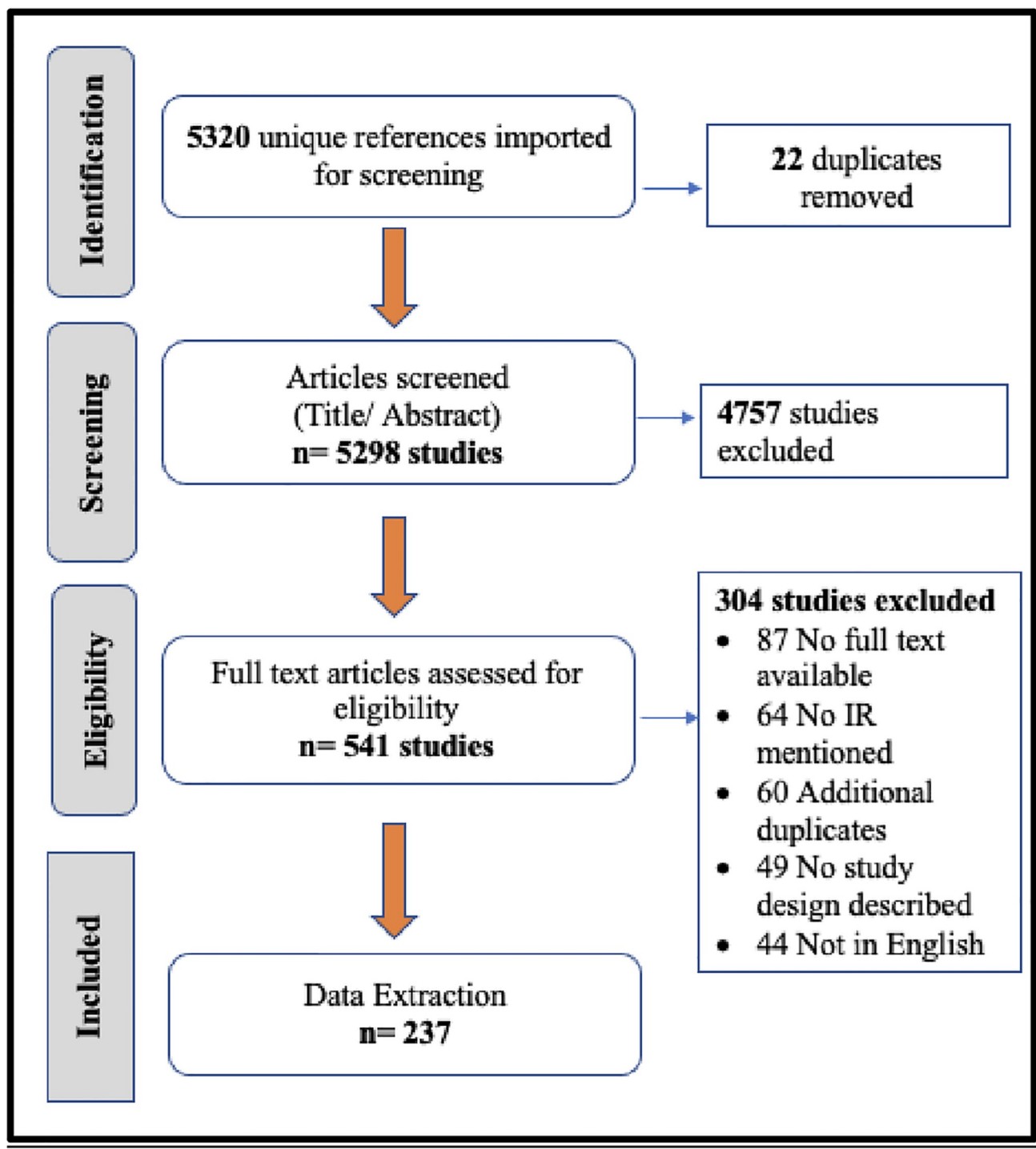

**Fig 1. PRISMA flow chart of study selection.** (Source: Authors).

## Inclusion and exclusion criteria

We included studies published between 1990–2021 if they contained information about the incidence of T1D in children, adolescents, and young adults, under 25 years of age. Data from

national diabetes registries or databases that explicitly tracked a pediatric or adolescent popula-tion were included. Studies were excluded if they did not report incidence rates of T1D or reported incidence rates for other diseases. Studies reporting incidence rates for populations over 25 years of age were excluded. Studies published in languages other than English were excluded from data extraction eligibility. Abstracts without associated full text, including abstracts published from conference presentations, were not included. Abstracts, where full text could not be located, were also excluded. All full-text manuscripts where the incidence rates were not reported were excluded from the review. Efforts were made to obtain the value of incidence rates for each country at the national level. If more than one study was available for a country, we applied the following four criteria to select the most suitable study: more recent studies, covering a large part of the country, including the age, ranges 0–14 and 15–19 years, providing age/sex-specific rates for 0–4, 5–9, 10–14 and 15–19-year age-groups.

## Screening, data extraction and analysis

A total of 7,647 results were retrieved across the three search databases (4,527 results from PubMed/ Medline; 2,484 results from Embase and 636 results from Global Health), resulting in 5320 unique references. Duplicate records were removed using EndNote and 22 additional duplicates were identified and removed during the import of references into the Covidence software for a title and abstract screening. 60 additional duplicates were discovered and excluded during full-text screening or data extraction. All abstracts were independently screened by three authors (AG, CR and MO) to identify articles meeting inclusion criteria. A process of consensus resolved disagreements, and inter-rater reliability was determined using a 20% random sample to calculate the Cohen's Kappa coefficient (k = 0.785). 538 full text arti-cles were reviewed by two authors for eligibility, with final inclusion of the 237 studies deter-mined by 2–3 author consensus. Data were extracted from the 237 studies by AG using a data collection template developed by ZW.

For each article, the following information was extracted and tabulated in the template (Table C in S1 Text): 1) Identification of the study with the year of publication: Authors, title, DOI and PMID; 2) Country and study location; 3) Geographical coverage of the study: Nation-wide (when the study was conducted across the whole nation) or regional (when the study was restricted to a region, city or single-center) or Global (when the study involved rates reported globally); 4) Incidence rates expressed as new cases per 100,000 people (for both sexes) per year in the following 5-year age/sex subgroups (males/females) 0–4, 5–9, 10–14, 15–19 years and also in 0–14 and 0–19 years, the age standard stratifications used by IDF, to ensure compa-rability of our results. The rates were retrieved from either the tables or graphs, with the study periods recorded for each study. Supplementary information was also reviewed to include information on incidence rates from the literature for reported study periods.

Data extraction was performed using Microsoft Excel 16.51, and statistical analysis was per-formed using R v3.3.61. Data were graphically represented on maps that contained informa-tion from countries at the national level. We compared the incidence of T1D for countries that had information for the age groups 0–19 years and 0–14 years. Regional means and 95% CIs were estimated using an inverse-variance weighting of the country-specific log incidence rates, re-transformed using Duan's smearing estimator [14]. Secular trends were estimated for each region by regressing the log incidence rate on the calendar year, with inverse-variance weight-ing used to weight each estimate. We performed subgroup analysis by assigning countries based on the World Health Organisation (WHO) regions and by country income groups based on gross national income (GNI) per capita in 2018, as published in the June 2019 World Bank Income Classification: low-income country (LIC) with per capita GNI of $1025, lower-

middle-income country (LMIC) $1026 to $3995, upper-middle-income country (UMIC) $3996 to $12,735, and high-income country (HIC) >$12,735 respectively [15].

## Results

### Study characteristics

We identified and extracted data from 237 studies, which provided 1852 estimates reporting T1D incidence covering 92 countries (Fig A in S1 Text). Of these studies, 79 reported T1D incidence in age groups 0–14 years, while 13 reported incidence in those aged 0–19 years. The majority of published studies (n = 1498, 81%) reported incidence estimates with accompanying 95% confidence intervals. For a minority of published studies (n = 354, 19%), estimated incidence rates of T1D were calculated by dividing the number of patients diagnosed with T1D (numerator) by the corresponding person-years at risk (denominator). Of the 1852 T1D incidence estimates presented, 50% (n = 920) were nationwide, 43% (n = 804) were regional or multi-center, and 7% (n = 128) were global (Fig B in S1 Text). Among the studies reporting incidence estimates from primary data sources, 64% (n = 1189) were reported from population-based registries (national or regional), and 36% (n = 663) were facility-based (single-center, hospital, or medical record) (Fig B in S1 Text).

### Variability in type 1 diabetes incidence rates

We observed substantial variation in the incidence of T1D worldwide and how it differed by age, sex, world region, and country income classification. We retrieved and compared national and subnational-level data from 92 countries in individuals 0–14 years and 0–19 years age, with sub-analysis for the IDF standard five-year age groups 0–4, 5–9, 10–14, and 15–19 years.

T1D incidence varied substantially between the 0–4, 5–9, 10–14, 15–19, 0–14 and 0–19 age groups across regions (Figs 2 and 3, Tables 1 and 2). Overall, the incidence was highest in the 10–14 year age group [18.02 per 100,000 (95% CI [17.54;21.49])] and lowest in the 15–19 year age group [6.71 per 100,000 (95% CI [4.54;7.91])]. Among those diagnosed in the 0–4 age category (Table 2), incidence was highest in Northern Africa [31.11 per 100,000 (95% CI [31.11;31.11])], Northern Europe [21.54 per 100,000 (95% CI [19.05;24.35]) and Western Europe [15.21 per 100,000 (95% CI [13.84;16.72]); it was lowest in Southern Asia, Eastern Asia, Western Asia and Eastern African countries (< 5 per 100,000), with a mean incidence of 10.27 per 100,000 (95% CI [8.77,12.97]). Within the 5–9 years age category (Table 2), incidence was highest in Northern Africa [44.78 per 100,000 (95% CI [NA])], Northern Europe [37.17 per 100,000 (95% CI [33.60,41.12])] and Northern America ([26.31 per 100,000 (95% CI [23.82,29.06])], whereas it was lowest in Southern Asia [0.92 per 100,000 (95% CI [0.65,1.31])], Eastern Asia [1.93 per 100,000 (95% CI [1.64,2.27])] and South America [4.47 per 100,000 (95% CI [1.51,13.25])], with a mean incidence of [17.19 per 100,000 (95% CI [15.67,19.41])]. Among those diagnosed within the 10–14 age category, incidence was highest in Northern Europe [41.48 per 100,000 (95% CI [37.42,45.97])], Northern Africa [40.92 per 100,000 (95% CI [NA])] and Northern America [33.50 per 100,000 (95% CI [29.54,38.00])], but lowest in Southern Asia [1.99 per 100,000 (95% CI [1.45,2.76])], Eastern Asia [2.78 per 100,000 (95% CI [2.41,3.21])] and South America [3.62 per 100,000 (95% CI [NA])] with a mean incidence of [18.02 per 100,000 (95% CI [17.54, 21.48])]. Finally, in the 15–19 age category (Table 2), incidence rates were highest in Northern America [17.68 per 100,000 (95% CI [13.31,23.48])] and Southern Europe [9.71 per 100,000 (95% CI [NA])], whereas they were lowest in East Asia [1.43 per 100,000 (95% CI [NA])] and African countries [1.07 per 100,000 (95% CI [0.58,1.96])], with a mean incidence of [6.71per 100,000 (95% CI [4.54, 7.91])].

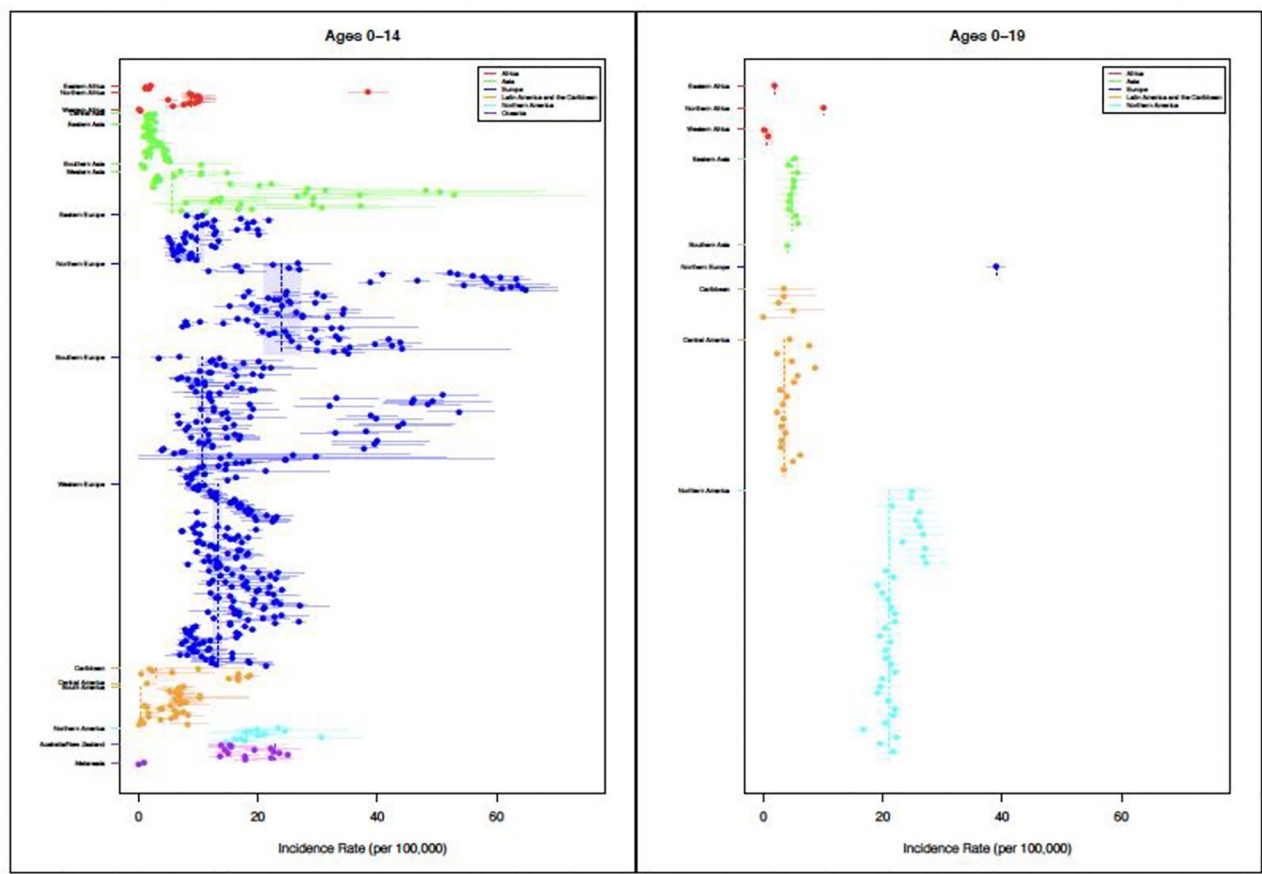

**Fig 2. Variation in TlD incidence rates for age groups 0–14 and 0–19 years by region.** [Point sizes are proportional to their weight (i.e. inverse variance). The lines indicate means and shaded areas indicate 95% Cls].

There was substantial variation in T1D incidence rates by country income group (Fig 4, Table 1). Higher incidence rates were reported in high-income countries [7.89 per 100,000 (95%CI [7.24;8.59])], followed by upper-middle-income [0.87 per 100,000 (95% CI [0.60;1.25])], lower-middle-income [0.57 per 100,000 (95% CI [0.34;0.98])] and low-income countries [0.19 per 100,000 (95% CI [0.13;0.28])]. Sub-analysis of incidence rates by sex (Table 3) shown as Fig C in S1 Text revealed a mean male-to-female ratio of 1.04 (95% CI [1.02, 1.06]).

Fig D in S1 Text depicts secular trends of T1D in different regions, with each circle proportionate to the weight of published literature reporting incidence rates. The overall incidence of T1D appears to be rising in some regions such as Australia, Northern America, and Europe, with relatively stable estimates reported from Asia. We noted very few older, and less reliable, estimates reported from Africa and Latin America dating back to the 1980s and 1990s, which hindered estimation of meaningful secular trends.

## Discussion

The results reveal substantial variability in T1D incidence worldwide by geographic region, country income classification, and age, which ranges from over 95% of cases in some regions (such as North America, Australia and Europe), to less than 35% in areas of Africa and Asia.

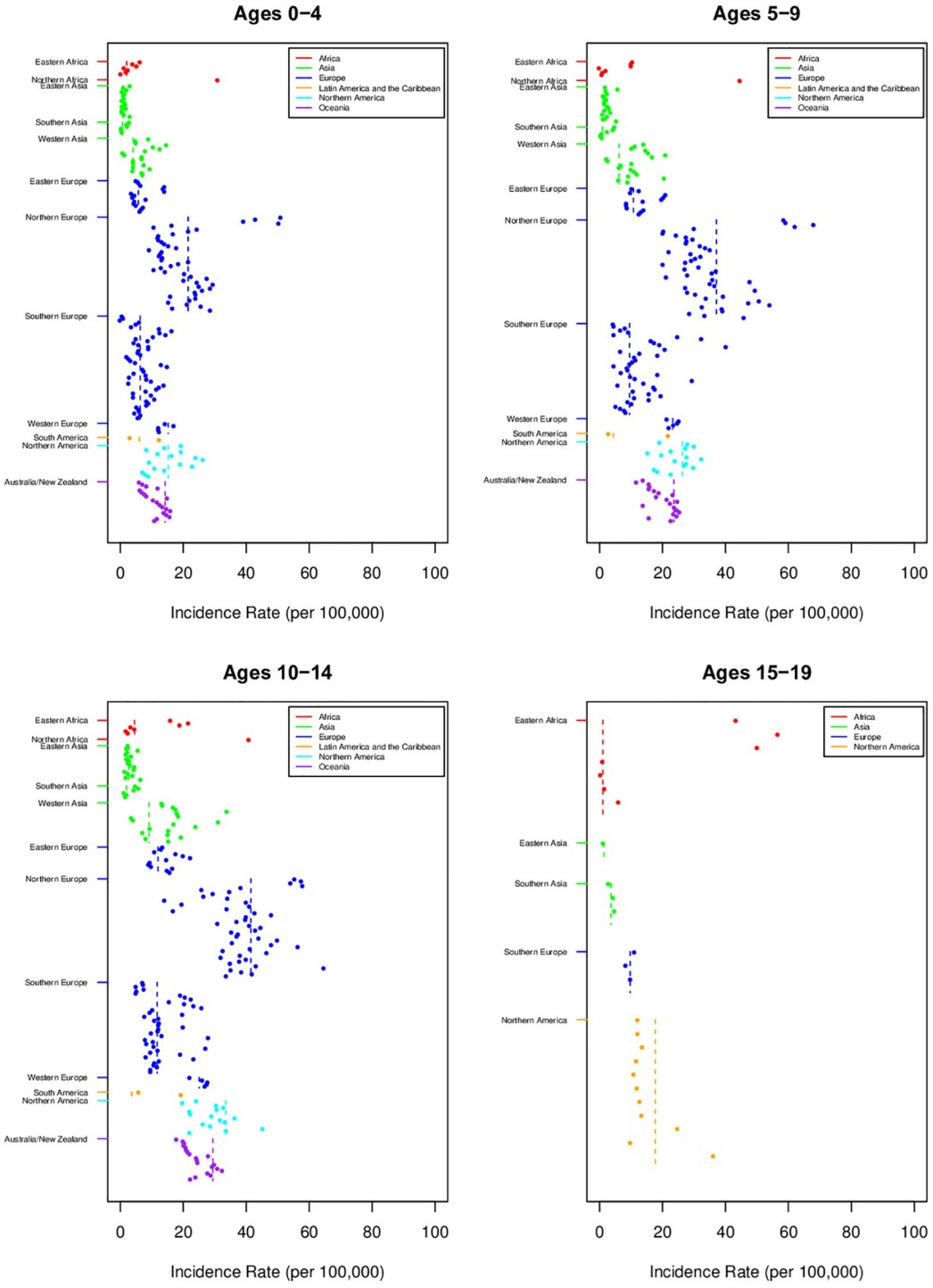

**Fig 3. Variation in T1D incidence rates for age groups 0–4, 5–9, 10–14, 15–19 years by region.** [Point sizes are proportional to their weight (i.e. inverse variance). The lines indicate means and shaded areas indicate 95% CIs].

**Table 1. Type 1 diabetes incidence rates in individuals aged 0–14 and 0–19 years by WHO regions and income.**
*(Age-standardised incidence rates per 100,000 individuals per year with 95% confidence intervals. † For cells labeled as NA, 95% CIs could not be estimated as there was only 1 data point).

| 0–14 years | | | |
|---|---|---|---|
| REGIONS | MEAN | 95% CI LOWER BOUND | 95% CI UPPER BOUND |
| Northern Europe | 23.95 | 21.11 | 27.18 |
| Australia/New Zealand | 22.80 | 20.97 | 24.79 |
| Northern America | 18.02 | 16.74 | 19.41 |
| Western Europe | 13.31 | 12.64 | 14.01 |
| Southern Europe | 10.79 | 10.02 | 11.63 |
| Eastern Europe | 10.00 | 9.04 | 11.07 |
| Northern Africa | 8.76 | 7.35 | 10.44 |
| Western Asia | 5.77 | 4.67 | 7.12 |
| Caribbean | 2.93 | 1.48 | 5.79 |
| Eastern Asia | 2.22 | 1.89 | 2.59 |
| Central Asia | 1.84 | 1.55 | 2.18 |
| Eastern Africa | 1.77 | 1.52 | 2.05 |
| Central America | 1.5 | NA[†] | NA[†] |
| Southern Asia | 0.69 | 0.49 | 0.98 |
| South America | 0.40 | 0.28 | 0.58 |
| Western Africa | 0.19 | 0.12 | 0.33 |
| Melanesia | 0.09 | 0.05 | 0.15 |
| **0–19 years** | | | |
| REGIONS | MEAN | 95% CI LOWER BOUND | 95% CI UPPER BOUND |
| Northern Europe | 39 | NA[†] | NA[†] |
| Northern America | 20.97 | 20.48 | 21.47 |
| Northern Africa | 10.1 | NA[†] | NA[†] |
| Eastern Asia | 4.79 | 4.46 | 5.16 |
| Southern Asia | 4 | NA[†] | NA[†] |
| Central America | 3.51 | 3.04 | 4.06 |
| Eastern Africa | 1.85 | NA | NA |
| Western Africa | 0.50 | 0.16 | 1.57 |
| Caribbean | NA[†] | NA[†] | NA[†] |
| INCOME GROUP | Mean | 95% CI LOWER BOUND | 95% CI UPPER BOUND |
| High income | 7.89 | 7.24 | 8.59 |
| Upper middle income | 0.87 | 0.60 | 1.25 |
| Lower middle income | 0.57 | 0.33 | 0.98 |
| Low income | 0.19 | 0.13 | 0.28 |

However, there is a paucity of country-level studies reporting data on incidence, especially in LMICs.

Worldwide, the incidence of T1D is increasing [16–20]. A recent study published global incidence and prevalence data of T1D using sensitivity analysis and estimated 234,710 and 9,004,610, respectively, in 2017 [21]. However, our review reveals more rigorous estimates and secular trends for each region supporting that increase in incidence and the rate of change is not uniform, with several notable nuances.

First, T1D incidence may follow a bimodal distribution pattern as regional differences are noted in peak incidence by age group. While we observed an increase in the incidence of T1D in the age categories of 5–9 years and 10–14 years, there appears to be substantial variation

**Table 2. Sub-analysis of type 1 diabetes incidence rates based on age category.** *(Age-standardised incidence rates per 100,000 individuals per year with 95% confidence intervals. † For cells labeled as NA, 95% CIs could not be estimated as there was only 1 data point).

| Region | Mean | 95% CI Lower Bound | 95% CI Upper Bound |
|---|---|---|---|
| **0–4 years** | | | |
| Northern Africa | 31.11 | 31.11 | 31.11 |
| Northern Europe | 21.54 | 19.05 | 24.35 |
| Western Europe | 15.21 | 13.84 | 16.72 |
| Northern America | 15.21 | 12.39 | 18.67 |
| Australia/New Zealand | 14.24 | 13.17 | 15.40 |
| Southern Europe | 6.36 | 4.05 | 9.98 |
| South America | 6.03 | 1.91 | 19.03 |
| Eastern Europe | 5.72 | 5.04 | 6.50 |
| Western Asia | 4.06 | 2.63 | 6.27 |
| Eastern Africa | 2.02 | 0.75 | 5.49 |
| Eastern Asia | 0.97 | 0.82 | 1.15 |
| Southern Asia | 0.70 | 0.51 | 0.96 |
| **5–9 years** | | | |
| Eastern Africa | NA[†] | NA[†] | NA[†] |
| Northern Africa | 44.78 | NA[†] | NA[†] |
| Northern Europe | 37.17 | 33.60 | 41.12 |
| Northern America | 26.31 | 23.82 | 29.06 |
| Australia/New Zealand | 23.64 | 22.37 | 24.99 |
| Western Europe | 23.27 | 21.68 | 24.98 |
| Eastern Europe | 10.79 | 9.51 | 12.25 |
| Southern Europe | 9.52 | 8.34 | 10.87 |
| Western Asia | 6.26 | 4.51 | 8.68 |
| South America | 4.47 | 1.51 | 13.25 |
| Eastern Asia | 1.93 | 1.64 | 2.27 |
| Southern Asia | 0.92 | 0.65 | 1.31 |
| **10–14 years** | | | |
| Northern Europe | 41.48 | 37.42 | 45.97 |
| Northern Africa | 40.92 | NA[†] | NA[†] |
| Northern America | 33.50 | 29.54 | 38.00 |
| Australia/New Zealand | 29.41 | 28.09 | 30.79 |
| Western Europe | 25.07 | 22.78 | 27.60 |
| Eastern Europe | 12.01 | 10.50 | 13.73 |
| Southern Europe | 11.72 | 10.51 | 13.08 |
| Western Asia | 9.12 | 6.71 | 12.40 |
| Eastern Africa | 4.54 | 2.62 | 7.87 |
| South America | 3.62 | NA[†] | NA[†] |
| Eastern Asia | 2.78 | 2.41 | 3.21 |
| Southern Asia | 1.99 | 1.45 | 2.76 |
| **15–19 years** | | | |
| Northern America | 17.68 | 13.31 | 23.48 |
| Southern Europe | 9.71 | NA[†] | NA[†] |
| Southern Asia | 3.67 | 2.83 | 4.77 |
| Eastern Asia | 1.43 | NA[†] | NA[†] |
| Eastern Africa | 1.07 | 0.58 | 1.96 |

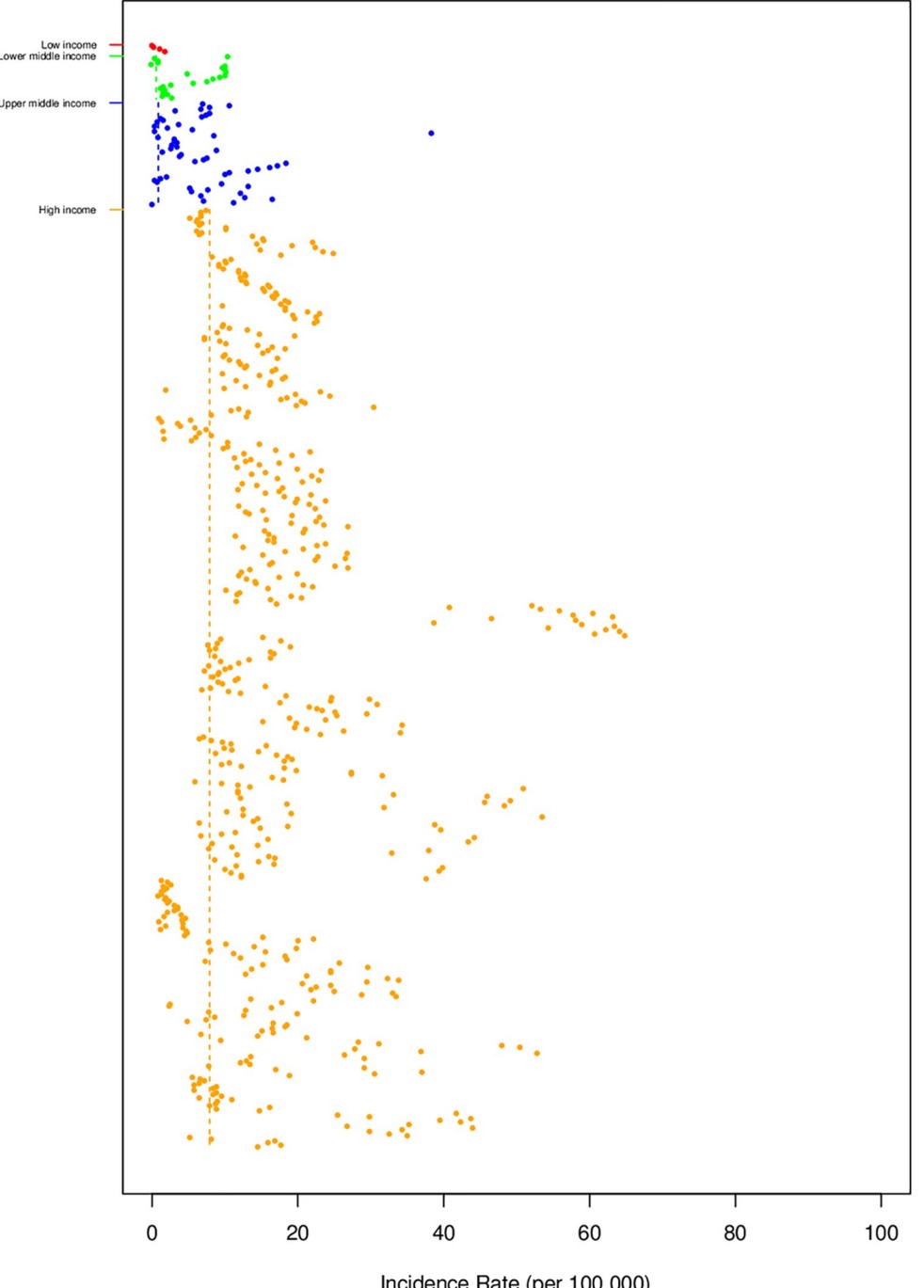

**Fig 4. Geographical representation of incidence rates of T1D by income.** [Point sizes are proportional to their weight (i.e. inverse variance). The lines indicate means and shaded areas indicate 95% CIs].

**Table 3. Sub analysis of type 1 diabetes incidence rates by sex.** *(Age-standardised incidence rates per 100,000 individuals per year with 95% confidence intervals. † For cells labeled as NA, 95% CIs could not be estimated as there was only 1 datapoint).

| Region | Mean | 95% CI Lower Bound | 95% CI Upper Bound |
|---|---|---|---|
| **Males** | | | |
| Northern Africa | 40.51 | NA† | NA† |
| Australia/New Zealand | 23.72 | 21.97 | 25.60 |
| Northern America | 18.57 | 14.49 | 23.79 |
| Northern Europe | 16.83 | 12.12 | 23.38 |
| Western Europe | 11.07 | 10.17 | 12.04 |
| Southern Europe | 10.96 | 10.01 | 11.99 |
| Western Asia | 10.93 | 10.11 | 11.80 |
| Eastern Europe | 8.64 | 7.72 | 9.68 |
| South America | 6.71 | 6.19 | 7.26 |
| Eastern Asia | 1.88 | 1.57 | 2.25 |
| Eastern Africa | 1.82 | NA† | NA† |
| Southern Asia | 1.01 | 0.95 | 1.07 |
| **Females** | | | |
| Northern Africa | 36.49 | NA† | NA† |
| Australia/New Zealand | 22.33 | 21.17 | 23.55 |
| Northern Europe | 18.29 | 13.27 | 25.23 |
| Northern America | 17.50 | 16.31 | 18.78 |
| Western Asia | 11.87 | 11.30 | 12.47 |
| Southern Europe | 10.05 | 9.26 | 10.90 |
| Western Europe | 10.00 | 9.16 | 10.91 |
| Eastern Europe | 8.66 | 7.71 | 9.72 |
| South America | 5.91 | 5.38 | 6.49 |
| Eastern Asia | 2.70 | 2.23 | 3.26 |
| Eastern Africa | 2.40 | NA† | NA† |
| Southern Asia | 1.05 | 0.92 | 1.20 |

when peaks occur in different populations. In Sweden, for example, the highest incidence was noted in the 0–9 years [27.1 (25.6–27.4)] [22], whereas, in a 14-year longitudinal study conducted in the United States, the highest incidence occurred in the 10–14 years (45.5/ 100,000 person-years) [23]. Interestingly, in the 0–4 years category, there appears to be less variability in incidence across world regions. One explanation for this observation could be the temporal relationship between T1D triggers (environmental, seasonal, socioeconomic background, and other contextual factors) and the onset of symptoms in at-risk individuals, in addition to differences in predisposition that vary within-country populations [16, 24–26].

Second, the incidence of T1D appears to be increasing with time. A recent study analyzing three distinct age categories in 15 countries over two time periods (1975–1999 and 2000–2017), reported an increased incidence in the 0–4 year age group (1.9 times), followed by the 5–9 year age group (1.8 times) and the 10–14 year age group (1.4 times) [27]. Epidemiological, environmental, and health system factors may explain this observation. More frequent and intense triggering events, reductions in competing childhood mortality with demographic transitions, and improved health system diagnostic capabilities in certain regions, among other factors, may all be contributory.

Third, there appears to be no difference in incidence of T1D by sex. However, there may be differences in the age of peak incidence by sex. The male to female incidence ratio was 1.04

(95% CI [1.02, 1.06]), which shows a slight predominance in males. Another study reported predominance in males by age ten and persisted throughout adulthood with the male to female incidence ratio of 1.32 (95% CI [1.30–1.35]) [23]. These sex differences may be explained with exposure to certain gendered behavioral practices and susceptibility to T1D environmental triggers or innate genetic predisposition and hormonal variance. Studies of sex differences in hormonal fluctuations at the time of adrenarche in genetically predisposed individuals may provide clues to the underlying pathogenesis of T1D [28, 29].

Fourth, context matters: there is a higher incidence of T1D observed in high-income settings. The incidence and prevalence are the highest in high-income countries which constitute only 10% of the world population in both the 0–14 and 0–19 year age groups [8]. The observed high incidence in these countries may be explained by health system factors and those relating to the social determinants of health. In high-income countries, more robust health systems enable more effective diagnosis of T1D and/or referral to specialists when the diagnosis is uncertain. In addition, patients that present with complications, for example, diabetes ketoacidosis (DKA), are more likely to receive timely and effective management needed for survival and diagnosis. Studies have also found that for some children an accurate T1D diagnosis is initially missed by clinicians even in some high-income settings [16, 30–33]. For example, a patient survey in the United States found that 25% of all patients reported being initially misdiagnosed with another condition, which was associated with 18% increased risk of DKA than those who were correctly diagnosed [34]. The potential barriers to receiving timely diagnosis as explored in the survey, included lack of a primary care provider, lack of time, lack of insurance, having a high deductible or copay, difficulty getting an appointment with a physician, and lack of transportation. In many LMICs, levels of misdiagnosis are likely higher, with studies finding that in sub-Saharan Africa most healthcare workers are not able to recognize symptoms and signs of DKA in a timely manner, which leads to high mortality [35, 36]. Indeed, many children presenting with diabetic coma in Africa are likely to be treated for more common diseases such as cerebral malaria or meningitis before the correct diagnosis is suspected [37]. It has therefore long been suspected that many children in such settings die before being correctly diagnosed [38, 39]. This helps to explain the paucity of data from many LMICs and emphasizes on the importance of accurate and timely diagnosis in these settings.

Higher incidence in high-income countries may also be explained by environmental and lifestyle or behavioral transitions in diet and physical activity associated with increases in GDP per capita. A recent study analyzing two study periods (1975–1999 and 2000–2017) found a positive correlation between T1D incidence and GDP per capita. By using a linear regression model, they suggested that for the year 1991, the country-to-country variation in GDP explained 9% of the country-to-country variation in incidence rates, which was 17% for the year 2006 [27]. This relationship with GDP could be explained by differences in behavior, lifestyle, and nutrition that are influenced by one's economic position. A positive relationship exists between a country's level of economic development as measured by GDP and obesity [40]. An elevated BMI and a sedentary lifestyle may exacerbate insulin resistance, which could lead to $\beta$-cells fatigue and triggering of an autoimmune response with resultant $\beta$-cell apoptosis [41, 42]. There exists a complex relationship between environmental factors and genetic risk for T1D or type 2 diabetes which likely plays a role in autoimmunity pathogenesis and presentation of clinical disease.

Globally, we find wide variability in incidence of T1D and higher estimates than recent estimates from the IDF Diabetes Atlas [1]. Several factors such as environmental triggers, ethnic differences, genetic susceptibility, and the ability of country health systems to diagnose new cases, are all logical explanations [6, 9, 12, 43, 44]. An association with the seasonality of onset has been reported in numerous studies in Northern Europe [45–47], among other countries

[44, 48, 49]. The cyclic, sinusoidal model of T1D "seasonality" has been reported, with a peak occurring in winter in both sexes in all age groups, but is more profound in regions with more significant temperature fluctuations [50]. In Norway and Finland, the incidence of T1D has decreased among young Finnish children, with findings implying that environmental factors driving the immune system toward islet autoimmunity are changing in young children [51, 52]. Differences in HLA association may alter predisposition to T1D, despite individuals being exposed to the same trigger. Health systems differ widely in their ability to effectively screen and diagnose individuals with diabetes, in addition to their capability to address other causes of competing mortality and prolong life that might otherwise have prevented the expression of T1D.

There is insufficient data available for Latin America and the Caribbean to estimate stable trends. While the literature reports low incidence rates in these regions, the frequency of diabetes in Latin America is expected to increase by 38% over the next ten years, compared with an estimated 14% increase in their total population, with overall numbers exceeding the number of cases in the US, Canada, and Europe by 2025 [53]. Recent evidence from Brazil demonstrates a marked increase in incidence rates of T1D (3.1% annually, with an absolute crude increase of 2.5-fold for the 0–14 year age group), making up almost three-quarters of total incidence in Latin America and the Caribbean [53].

The studies from Africa reported notably higher incidence rates in the 15–19 year age group compared to the 0–14 year age group, which may reveal different patterns of incidence and drivers of T1D by age-group and region or simply that weak health systems and access to healthcare matter even more. Simulation-based analysis estimates that by 2050, the total number of new cases of T1D are projected to increase especially in Africa which will account for 51% of global new cases of T1D per year and highest DKA admissions at diagnosis [54].

Further reliable and recent research from the sub-Saharan Africa region will help to refine this ratio. Africa has a rapidly growing population of children, adolescents, and young adults, and knowledge of the recent trends is crucial to develop appropriate policies and healthcare services that improve health outcomes.

Our review reveals the paucity of data available on the incidence of T1D among children and adolescents in most world regions and the need to redouble efforts on developing efficient data systems to collect, pool, and store reliable healthcare data, particularly in LMICs. While the overall impression is wide heterogeneity in the incidence rates of T1D and plateauing in the Scandinavian countries, as data systems are strengthened, the true nature of the incidence dynamics will be revealed. National population-based prospective registries provide a platform to obtain estimated T1D incidence rates. However, such initiatives are typically only sustained in high-income countries leading to a discrepancy in data and incidence estimates at the global level. Investing in data systems to capture data more efficiently, especially in Africa, where investments are needed to strengthen health systems that could inform policy and practices related to resource allocation and the development of targeted interventions to improve the effectiveness, equity, efficiency, and responsiveness of healthcare services for children and adolescents with T1D. One such example could be diagonally integrating diabetes care into the broader health system in order to improve the quality of care for T1D and also other chronic illnesses [55–57]. This could include policy prioritization, innovative financing, task-shifting, secure and sustainable access to essential medicines and diagnostic tools, and comprehensive service delivery.

There were several limitations to this study. First, we were not able to obtain information for crucial associations including socioeconomic status (for example, education level and occupational status, among other proxy measures), health-seeking behavior, ethnicity, and seasonal variability due to the incompleteness of data from all 237 included studies. When reporting the

regional incidence rates, we did not control for these crucial associations. Second, inclusion of both population-based and cohort studies could serve as a bias and may not truly represent the population characteristics in each country. In addition, studies that were published in non-English languages such as Spanish, Russian and French were also not included in our investigation due to time constraints and the non-availability of an English translation. Third, the low accuracy of decrease in incidence trends with age beyond 14 years was not extensively explored due to the paucity of data available for the age category of 15–19 years. Finally, this study was unable to obtain data on the trends of DKA and mortality rates which warrants further investigations in future reviews.

Notwithstanding these limitations, this first global scoping review reveals the substantial variations in the incidence rates of T1D worldwide by region, country income group, and age category. There is a substantial paucity of data from LMICs and comparatively overwhelming evidence from high-income countries. A more accurate and holistic picture of the global burden of T1D is critical to inform health policies to strengthen health systems and improve access to effective, efficient, equitable and responsive healthcare services for children and adolescents and improve health outcomes.

## Supporting information

**S1 Text.**
(PDF)

## Author Contributions

**Conceptualization:** Ché L. Reddy, Rifat Atun.

**Data curation:** Apoorva Gomber, Zachary J. Ward, Ché L. Reddy, Rifat Atun.

**Formal analysis:** Apoorva Gomber, Zachary J. Ward.

**Investigation:** Apoorva Gomber.

**Methodology:** Apoorva Gomber, Zachary J. Ward, Carol Mita.

**Project administration:** Rifat Atun.

**Resources:** Apoorva Gomber, Carlo Ross.

**Software:** Apoorva Gomber.

**Supervision:** Ché L. Reddy, Rifat Atun.

**Validation:** Apoorva Gomber, Carlo Ross, Maira Owais, Ché L. Reddy, Rifat Atun.

**Visualization:** Apoorva Gomber, Carlo Ross, Maira Owais, Ché L. Reddy, Rifat Atun.

**Writing – original draft:** Apoorva Gomber.

**Writing – review & editing:** Apoorva Gomber, Zachary J. Ward, Carlo Ross, Maira Owais, Carol Mita, Jennifer M. Yeh, Ché L. Reddy, Rifat Atun.

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
