## [Decision Letter · Decision Letter 0]

5 Jul 2022

PGPH-D-22-00825

Variation in the Incidence of Type 1 Diabetes Mellitus In Children and Adolescents by World Region and Country Income Group: A Scoping Review

Dear Author,

Thank you for submitting your manuscript to PLOS Global Public Health. After careful consideration, we feel that it has merit but does not fully meet PLOS Global Public Health’s publication criteria as it currently stands. Therefore, we invite you to submit a revised version of the manuscript that addresses the points raised during the review process.

Please review the feedback from the reviewers.  The revisions are minor and we welcome a resubmission of your manscript.

Revisions required:

On page 16, paragraph 1, the authors start the sentence by “What explains the wider variability…”- while there is no harm starting the paragraph with a question, the sentence structure here is poorly done. Authors may need to restructure.

Also, while authors mention that access to healthcare in high income countries is key in timely diagnosis of T1D, and that strengthened healthcare system etc is important in ensuring access to data, authors need to expound further why lack of these two is an impediment in getting accurate or timely information on the T1D incidence in many LMICs.

I like how the author speculate that it may be possible that many children die earlier in SSA before diagnosis. Can authors build up on this and add relevant references.

Also, authors have associated lifestyle changes and obesity cases with T1D in developed countries. However, we know that under nutrition as well as eating unhealthy diets and physical inactivity are major issues contributing to NCDs in LMIC today. Can the author substantiate on this by comparing developed and developing countries?

Authors have mentioned few policy implications for this work. I feel this study is very important and authors should add a paragraph or so on the research implications especially in African countries. Also, we know that LMIcs are lagging behind in terms of research, databases, and data sharing principles. What need to be done?

We look forward to receiving your revised manuscript.

Kind regards,

Kate Frazer

Academic Editor

Journal Requirements:

1. Please amend your online Financial Disclosure statement. If you did not receive any funding for this study, please simply state: “The authors received no specific funding for this work.”

2. Please update your online Competing Interests statement. If you have no competing interests to declare, please state: “The authors have declared that no competing interests exist.”

3. Please provide a complete Data Availability Statement in the submission form, ensuring you include all necessary access information or a reason for why you are unable to make your data freely accessible. If your research concerns only data provided within your submission, please write “All data are in the manuscript and/or supporting information files.” as your Data Availability Statement.

4. We have noticed that you have uploaded Supporting Information files, but you have not included a list of legends. Please add a full list of legends for your Supporting Information files after the references list.

5. All figures and supporting information files will be published under the Creative Commons Attribution License (creativecommons.org/licenses/by/4.0/). Authors retain ownership of the copyright for their article and are responsible for third-party content used in the article. 

Figure 1 (Supplementary Figure): please (a) provide a direct link to the base layer of the map used and ensure this is also included in the figure legend; (b) provide a link to the terms of use / license information for the base layer. We cannot publish proprietary or copyrighted maps (e.g. Google Maps, Mapquest) and the terms of use for your map base layer must be compatible with our CC-BY 4.0 license. 

Please upload any written confirmation as an 'Other' file type. It must clarify that the copyright holder understands and agrees to the terms of the CC BY 4.0 license; general permission forms that do not specify permission to publish under the CC BY 4.0 will not be accepted. Note that uploading an email confirmation is acceptable.

Additional Editor Comments (if provided):

Reviewers' comments:

Reviewer's Responses to Questions

**Comments to the Author**

1. Does this manuscript meet PLOS Global Public Health’s publication criteria? Is the manuscript technically sound, and do the data support the conclusions? The manuscript must describe methodologically and ethically rigorous research with conclusions that are appropriately drawn based on the data presented.

Reviewer #1: Yes

Reviewer #2: Yes

2. Has the statistical analysis been performed appropriately and rigorously?

Reviewer #1: Yes

Reviewer #2: Yes

3. Have the authors made all data underlying the findings in their manuscript fully available (please refer to the Data Availability Statement at the start of the manuscript PDF file)?

Reviewer #1: No

Reviewer #2: Yes

4. Is the manuscript presented in an intelligible fashion and written in standard English?

Reviewer #1: Yes

Reviewer #2: Yes

5. Review Comments to the Author

Reviewer #1: Dear Authors,

This study is a scoping literature review on the incidence of type 1 diabetes in children, adolescents, and young adults (0-25 years) worldwide and explains the wide heterogeneity in incidence rates that differ by age, sex, world region, and country income classification. I took keen interest to read through this important study , which I find to be timely especially given that incidences of T1D globally and in SSA are not well documented. The authors did a great job in categorizing their analysis and results in terms of region, age etc and also used the IDF age classification to conduct their analysis. Findings from this study are important not only to readers of Plos GPH, but also scientists, policy makers and other stakeholders interested in the field of diabetes broadly and T1D specifically.

The introduction section is well summarized and the gap is well stated. However, authors should ensure that they include relevant references where necessary. For example, paragraph 1, after LMIC, authors should add a reference.

The methodology section is well explained. The authors have followed key steps of scoping review and processes of the scoping review are well presented. This study can easily be replicated by other researchers.

Results and discussion: these sections are presented in a logical fashion. The presentation is also easier to understand. Statistics and presentations are well done .

Overall, the writing style is good and in standard English. The language is consistent, short and meaningful paragraphs that makes it easy for the reader to follow the narrative.

I have few things that the may consider correcting to strengthen their manuscript:

On page 16, paragraph 1, the authors start the sentence by “What explains the wider variability…”- while there is no harm starting the paragraph with a question, the sentence structure here is poorly done. Authors may need to restructure.

Also, while authors mention that access to healthcare in high income countries is key in timely diagnosis of T1D, and that strengthened healthcare system etc is important in ensuring access to data, authors need to expound further why lack of these two is an impediment in getting accurate or timely information on the T1D incidence in many LMICs. I like how the author speculate that it may be possible that many children die earlier in SSA before diagnosis. Can authors build up on this and add relevant references.

Also, authors have associated lifestyle changes and obesity cases with T1D in developed countries. However, we know that under nutrition as well as eating unhealthy diets and physical inactivity are major issues contributing to NCDs in LMICs today. Can the author substantiate on this by comparing developed and developing countries?

Authors have mentioned few policy implications for this work. I feel this study is very important and authors should add a paragraph or so on the research implications especially in African countries. Also, we know that LMICs are lagging behind in terms of research, databases, and data sharing principles. What need to be done?

Reviewer #2: I thank the authors for sharing their work for publication.

In the present study, the authors have attempted to assess the Type 1 Diabetes Mellitus Incidence in Children and Adolescents Variation by World Region and Country Income Group. The review concludes T1D incidence rates vary significantly by geographic region, country economic level, and age.

Overall the study offers an important piece of information.

1. The title properly reflects the subject of the paper.

2. The abstract is short, accurate and clear. It provides an accessible summary of the paper.

3. The keywords accurately reflect the content.

4. Introduction is well written and Summarizes recent research related to the topic

5. The methods are replicable and follows the best practice standards

6. The result seems plausible and the study provides sufficient data, clear data tables and non-contradictory data that agree with the conclusions.

7. The referencing is accurate, adequate and balanced.

6. PLOS authors have the option to publish the peer review history of their article (what does this mean?). If published, this will include your full peer review and any attached files.

**Do you want your identity to be public for this peer review?** For information about this choice, including consent withdrawal, please see our Privacy Policy.

Reviewer #1: No

Reviewer #2: No

---

## [Decision Letter · Decision Letter 1]

27 Sep 2022

PGPH-D-22-00825R1

Variation in the Incidence of Type 1 Diabetes Mellitus In Children and Adolescents by World Region and Country Income Group: A Scoping Review

Dear Dr. Gomber,

Thank you for submitting your manuscript to PLOS Global Public Health. After careful consideration, we feel that it has merit but does not fully meet PLOS Global Public Health’s publication criteria as it currently stands. Therefore, we invite you to submit a revised version of the manuscript that addresses the points raised during the review process.

Please submit your revised manuscript by October 11 of 2022. If you will need more time than this to complete your revisions, please reply to this message or contact the journal office at globalpubhealth@plos.org. Please include the following items when submitting your revised manuscript:

We look forward to receiving your revised manuscript.

Kind regards,

Claudio A. Mendez, MPH

Academic Editor

Journal Requirements:

Additional Editor Comments (if provided):

Reviewers' comments:

Reviewer's Responses to Questions

**Comments to the Author**

1. If the authors have adequately addressed your comments raised in a previous round of review and you feel that this manuscript is now acceptable for publication, you may indicate that here to bypass the “Comments to the Author” section, enter your conflict of interest statement in the “Confidential to Editor” section, and submit your "Accept" recommendation.

Reviewer #1: All comments have been addressed

Reviewer #2: All comments have been addressed

2. Does this manuscript meet PLOS Global Public Health’s publication criteria? Is the manuscript technically sound, and do the data support the conclusions? The manuscript must describe methodologically and ethically rigorous research with conclusions that are appropriately drawn based on the data presented.

Reviewer #1: Yes

Reviewer #2: Yes

3. Has the statistical analysis been performed appropriately and rigorously?

Reviewer #1: Yes

Reviewer #2: Yes

4. Have the authors made all data underlying the findings in their manuscript fully available (please refer to the Data Availability Statement at the start of the manuscript PDF file)?

Reviewer #1: Yes

Reviewer #2: Yes

5. Is the manuscript presented in an intelligible fashion and written in standard English?

Reviewer #1: Yes

Reviewer #2: Yes

6. Review Comments to the Author

Reviewer #1: The authors have addressed all my questions. The manuscripts is well written and additional information added in text has made the manuscript stronger. Well done to this important work.

Reviewer #2: The manuscript has been improved, and all comments have been taken into account. I send my best wishes for publication.

7. PLOS authors have the option to publish the peer review history of their article (what does this mean?). If published, this will include your full peer review and any attached files.

**Do you want your identity to be public for this peer review?** For information about this choice, including consent withdrawal, please see our Privacy Policy.

Reviewer #1: **Yes: **Edna N Bosire

Reviewer #2: No

---

## [Editor Report · Decision Letter 2]

10 Oct 2022

Variation in the Incidence of Type 1 Diabetes Mellitus In Children and Adolescents by World Region and Country Income Group: A Scoping Review

PGPH-D-22-00825R2

Dear Dr. Gomber,

We are pleased to inform you that your manuscript 'Variation in the Incidence of Type 1 Diabetes Mellitus In Children and Adolescents by World Region and Country Income Group: A Scoping Review' has been provisionally accepted for publication in PLOS Global Public Health.

Best regards,

Claudio A. Mendez, MPH

Academic Editor
